# Zinc Nutritional Status and Risk Factors of Elderly in the China Adult Chronic Disease and Nutrition Surveillance 2015

**DOI:** 10.3390/nu13093086

**Published:** 2021-09-02

**Authors:** Jiaxi Lu, Yichun Hu, Min Li, Xiaobing Liu, Rui Wang, Deqian Mao, Xiaoguang Yang, Lichen Yang

**Affiliations:** Key Laboratory of Trace Element Nutrition, National Health Commission of the People’s Republic of China, National Institute for Nutrition and Health, Chinese Center for Disease Control and Prevention, Beijing 100050, China; lujx@ninh.chinacdc.cn (J.L.); huyc@ninh.chinacdc.cn (Y.H.); limin@ninh.chinacdc.cn (M.L.); liuxb@ninh.chinacdc.cn (X.L.); wangr@ninh.chinacdc.cn (R.W.); maodq@ninh.chinacdc.cn (D.M.); xgyangcdc@163.com (X.Y.)

**Keywords:** serum zinc, zinc deficiency, Chinese elderly

## Abstract

Objectives: To analyze the serum zinc nutrition status of the Chinese elderly, and to assess the risk factor for zinc deficiency. Methods: 3727 elderly people over 60 years old were randomly selected from 302 monitoring points in 31 provinces of China Adult Chronic Disease and Nutrition Surveillance (CACDNS) 2015. Blood samples were selected from the biological sample bank of CACDNS 2015 and the basic information were collected by questionnaires. The criteria of serum zinc deficiency recommended by the International Zinc Nutrition Consulting Group (IZiNCG) were adopted, and the related factors were also analyzed. Results: The median serum zinc concentration of Chinese elderly was 99.2 (84.3~118.7) μg/dL. The overall adjusted prevalence of zinc deficiency was 8.68%, with a 95% confidence interval (CI) of 7.74% and 9.61%. Significant differences were found in different sex, age groups, Body Mass Index (BMI), residence region, race and education level in terms of serum zinc status in the elderly (*p* < 0.05). Participants who are males, aged above 70y, with BMIs of less than 18.5 kg/m^2^, living in rural areas, minority, and with the lowest income had a higher prevalence of zinc deficiency in the subgroups under different classifications. The risk of zinc deficiency in the elderly over 70y was 1.44 higher than those aged 60–69y (OR = 1.44, 95%CI 1.14–1.82), and the minority elderly has a 1.39 higher risk than Han nationality (OR = 1.39, 95%CI 1.01–1.91), while overweight/obesity and female elderly were at lower risk (OR < 1, *p* < 0.05). Conclusions: The overall prevalence of zinc deficiency in the elderly was highest in all adults in the CACDNS. It is recommended that the male elderly, over 70 years, and the minority elderly should pay more attention to the zinc nutrition status of their own.

## 1. Introduction

Zinc is one of the essential micronutrients for the human body. It participates in the metabolic process of more than 300 enzymes in the body. It can promote growth and development, improve immunity, maintain taste balance, ensure the nutritional flora of the gastrointestinal tract is in a normal state, and maintain the normal metabolic function of the liver [1,2]. Zinc deficiency occurs when the absorption of zinc cannot meet physiological needs. It is estimated that about 20% of the global population is at risk of zinc deficiency [3], threatening the health of children, women in pregnancy and lactation, and the elderly.

With the development of social economy and politics, life expectancy is increasing, and the global population has accelerated into an aging society. By the end of 2019, the number of elderly people over 65 years old in China had reached 176 million [4]. Due to the decrease in appetite and food intake and the corresponding decrease in nutrients intake through diet [5], the elderly are more likely to suffer from malnutrition, including micronutrient deficiency [6]. Zinc deficiency in the elderly affects immune function, cognitive ability [7], taste and many other aspects of health problems.

So far, there is no national representative report on the zinc nutrition status of the elderly in China. The aim of the study is to assess the zinc nutrition status of the elderly in China. We analyzed the serum zinc level for the first time based on the biological sample bank of China Adult Chronic Disease and Nutrition Surveillance (CACDNS) 2015, and we also assessed the related risk factors for zinc deficiency.

## 2. Materials and Methods

### 2.1. Study Population

This study was a cross-sectional study. Data were obtained from the CACDNS 2015. The 31 provinces (autonomous regions and municipalities, except Taiwan, Hong Kong, and Macau) across the country are divided into 8 levels according to the urbanization rate (high, low), population (high, low), and mortality rate (high, low), and finally, 302 monitoring points are selected. Please refer to the literature [8] for the sampling of specific plans. In this study, a simple random sampling method was used to select survey subjects at each monitoring site. Calculated by the formula of cross-sectional sample demand, a total of 3624 samples are required for this study. Taking into account the possible lack of samples due to exclusion criteria, an average of 14 samples for each monitoring point was determined, half male and half female. Exclusion criteria: incomplete questionnaire information, poor blood sample quality (hemolysis or insufficient sample size) and below the detection limit. Finally, 3727 elderly people over 60y were included. All the subjects signed an informed consent form. This project was approved by the Ethical Review Committee of the Chinese Center for Disease Control and Prevention (Lot Number: 201519-A).

### 2.2. Basic Information and Blood Sample Collection

All the basic information were recorded by questionnaire, including sex, age, BMI, residence region (urban and rural), location (eastern, midlands, and western) regions, race, education level, family income per annum. Among them, BMI data are calculated by height and weight. The height and weight of the subjects were measured with tools of uniform brands and models. The division of eastern, central and western regions is based on the “2018 China Health Statistics Yearbook” [9]. An amount of 8.0 mL of fasting venous blood of the subject was collected at the survey site, placed in the dark for 30 min, centrifuged at 1500 rpm for 15 min to separate the serum, and stored in a refrigerator at −20 °C. Then, it was transported back to the laboratory through the cold chain and stored in a refrigerator at −70 °C before testing.

### 2.3. Serum Zinc and Hemoglobin (Hb) Laboratory Analyses and Evaluation Standards

Before analysis, the frozen serum was thawed naturally at room temperature. An amount of 100 μL serum was absorbed in a 15 mL centrifuge tube, diluted 20 by times with 0.5% (*v/v*) high-purity dilute nitric acid to 2 mL, and mixed well before testing. The serum zinc concentration was detected by the inductively coupled plasma mass spectrometry (ICP-MS) method. In order to monitor the stability and analytical accuracy of the testing, high and low levels of commercial quality control (Sero norm™, Level-1, Level-2, Olso, Norway) were detected every 20 serum samples during the whole testing process. The results of the quality control test showed that the daily measured values were within the scope of quality control. The recoveries ranged from 89.5% to 96.6%, the inter-day accuracy was 2.3%, and the intra-day accuracy was 2.7–8.4%.

We adopted the criteria for determining serum zinc deficiency recommended by the International Zinc Nutrition Consulting Group (IZiNCG) [10], and the cut-off value for elderly men and women was serum zinc < 74 μg/dL and <70 μg/dL, respectively.

The fingertip blood of the subjects was taken at the scene, Hb level was determined by the cyanmethemoglobin method. According to the World Health Organization (WHO) anemia diagnostic criteria, anemia is defined as Hb < 130 g/L in elderly men and Hb < 120 g/L in elderly women at sea level. The anemia diagnosis standard [11] was adjusted by the altitude of each monitoring location.

### 2.4. Statistical Analyses

SAS 9.4 (SAS Institute Inc., Cary, North Carolina, USA) was used for all data cleaning and analysis. Serum zinc concentration was finally expressed as median and interquartile interval (IQR) for abnormal distribution, and the difference between the groups is tested by Kruskal–Wallis. Taking the sixth national census in 2010 as the standard population by age, sex, and urban and rural population, the weighted adjustments were performed afterwards, the chi-square test was used to compare the serum zinc deficiency rate of the elderly with different characteristics. The factors influencing the elderly zinc nutrition were analyzed by the multivariate logistic regression model. *p* < 0.05 indicates that the difference is statistically significant.

## 3. Results

### 3.1. Serum Zinc Status of Elderly

In this study, a total of 3727 elderly were tested for serum zinc concentration, the age was 71.8 ± 7.5 years, BMI was 23.6 ± 3.6 kg/m^2^. Zinc nutritional status of the elderly in average CACDNS 2015 was shown in Table 1. The median of serum zinc concentration in the elderly was 99.2 μg/dL, with an IQR of 84.3 μg/dL and 118.7 μg/dL. The serum zinc concentration of the elderly is related to sex, age group, BMI, residence region, race and education level. Serum zinc concentration in elderly men is higher than in elderly women (*p* < 0.05), the 60–69 years old elderly have the highest serum zinc concentration among all the elderly (*p* < 0.05), and the serum zinc concentration of elderly with overweight is 103.0 μg/dL, higher than low and normal weight elderly (*p* < 0.05). The serum zinc concentration of the elderly living in the city is higher than that of the rural elderly (*p* < 0.05), and Han elderly serum zinc concentration was 99.8 μg/dL, which is higher than that of the ethnic minority elderly (*p* < 0.05). Zinc level in elderly people with elementary school education and below was lowest (*p* < 0.05). There was no significant difference in serum zinc concentration between the elderly in the eastern, midlands and western regions and the elderly with different family incomes (*p* > 0.05).

### 3.2. Prevalence of Zinc Deficiency and Multivariate Logistic Regression Analysis

The overall prevalence of serum zinc deficiency was 8.68%, with a 95% CI of 7.74% and 9.61% (Table 2). The serum zinc deficiency rate of the elderly men, minorities, over 70 years old, low and normal weight, lived in rural areas and with an annual family income less than 2500 yuan are relatively higher in each subgroup (*p* < 0.05). There was no significant difference in the serum zinc deficiency rate in anemia, different location and education level (*p* < 0.05).

The risk factors of zinc deficiency rate were further explored by multivariate logistic regression analysis. It can be seen from Table 2 that the OR value of serum zinc deficiency in elderly women was 0.54 compared with that in elderly men; compared with 60–69 years elderly, the OR value of serum zinc deficiency in people over 70 years old was 1.44 (*p* < 0.05); for the elderly with normal weight, the OR values of overweight and obesity were 0.67 and 0.52, respectively (*p* < 0.05); compared with Han elderly, serum zinc deficiency in the minority elderly OR = 1.39 (*p* < 0.05).

## 4. Discussion

Zinc deficiency is still a major public health problem that significantly impacts developing countries [12]. This study aims to evaluate the zinc nutritional status of the elderly in CACDNS 2015 and analyze the related influencing factors.

At present, there are different indexes for evaluating zinc nutritional status, such as whole blood [13,14], or zinc contents in hair, urine and nails [2]. Among them, the serum zinc concentration is the most widely used indicator, and it is adopted by many international organizations, including IZiNCG, WHO, United Nations International Children’s Emergency Fund (UNICEF), International Atomic Energy Agency (IAES). The serum zinc concentration can well reflect the nutritional supplement effect of dietary zinc and zinc supplements, and is suitable for evaluating the zinc nutritional status of all populations [10].

IZiNCG recommends that if more than 20% of the population’s serum zinc concentration is below the critical value, it is considered that the population’s risk of zinc deficiency is increased [15]. There is little related literature on the zinc nutrition status of the elderly in China. In the present study, the zinc deficiency rate of the elderly in CACDNS 2015 was 8.68%, which was quite lower than the recommended 20% and was also lower than the 12.2% prevalence for 131 middle-aged and elderly people living in Wuhan reported in 2010 [14]. However, when comparing with the other population in CACDNS 2015, we found that the zinc deficiency rate in elderly were higher than the adult aged 18–60y (6.04%) [16], the childbearing women (5.03%) [17] and the pregnant women (3.5%) [18]. Therefore, it is still necessary to continue to pay more attention to this population. 

This study found that males, aged above 70y, and minorities were risk factors for zinc deficiency. Generally, the physiological requirement of zinc in males is higher than in females. The Dietary Reference Intakes (DRIs) in China [19] also recommended a higher dietary reference intake for males than for females. In our study, the fact that male elderly were at higher risk than females might be related to this. The decrease in food intake, chewing and digestive capacity with age may explain why the elderly over 70y are more at risk of zinc deficiency [5]. The rate of zinc deficiency in the elderly of ethnic minorities is higher than that of the Han ethnic group, and that in the rural elderly was also higher than the urban, which may be caused by different dietary habits and food supply [20].

On the other hand, we found overweight/obesity might be protective factors for zinc deficiency. This result was consistent with another report about the 18–60 years adults [16]. Zinc is mainly found in foods such as meat, seafood, and nuts. Obese and overweight people are more likely to consume more such foods than normal and low weight people [21]. This might explain the lower zinc deficiency in the overweight/obesity elderly.

The anemia rate in the subjects of our study was 10%, which was highest in all the adults recruited in CACDNS 2015 (data unpublished). It was reported recently that zinc concentrations are independently and positively associated with Hb concentrations in preschool children and women of reproductive age [22]. In our study, the results reveal that the elderly with anemia showed a slightly higher zinc deficiency rate, but the difference was not statistical (*p* > 0.05). Moreover, the result of the logistic regression analysis also exhibited that anemia was not the risk factor for zinc deficiency in the elderly (OR = 1.01, 95%CI 0.69–1.46). More research is warranted in this field.

The last thing to mention is the zinc deficiency criterion used in this study, which was recommended by IZiNCG according to some data from National Health and Nutrition Examination Survey data in the US. In order to more accurately evaluate the serum zinc nutritional status of the elderly in China, it is necessary to establish the optimal threshold based on Chinese surveillance data. Our team published the appropriate range of zinc in the blood of Chinese women of childbearing age [17]. In the next step, we will also study the normal serum zinc distribution standard of Chinese elderly people.

## 5. Conclusions

This is the first nationally representative survey data about zinc status in the Chinese elderly. We found that even though the overall prevalence of zinc deficiency in the Chinese elderly was not prominent (8.68%), it was still higher than the other population in CACDNS 2015. It was suggested that the male elderly, over 70 years, and the minority old should pay more attention to the monitoring of zinc nutrition since they were at higher risk.

## Figures and Tables

**Table 1 nutrients-13-03086-t001:** Distribution of serum zinc of the elderly in CACDNS 2015.

Variables	N (Percent %)	Median (μg/dL)	IQR (μg/dL)	χ^2^ Value	*p* Value
Total population	3727	99.2	84.3~118.7		
Sex				4.76	0.029
Man	1894(50.8)	98.7	83.3~117.6		
Woman	1833(49.2)	100.0	85.8~119.6		
Age group (Years)				26.24	<0.001
60~69	1635(43.9)	101.8 ^a^	86.7~122.0		
70~79	1559(41.8)	97.6 ^b^	83.1~116.7		
≥80	533(14.3)	97.0 ^b^	80.8~113.4		
BMI(kg/m^2^) *				36.41	<0.001
low weight (<18.5)	252(6.8)	96.7 ^a^	80.4~120.0		
normal (18.5~<24.0)	1890(50.7)	96.6 ^a^	82.0~116.6		
overweight (24.0~<28.0)	1147(30.8)	103.0 ^b^	88.0~122.0		
obesity(≥28.0)	438(11.7)	100.4 ^ab^	86.2~116.9		
Anemia				1.36	0.24
No	3324(90.1)	99.3	84.5~111.9		
Yes	367(9.9)	99.1	82.2~116.7		
Residence region				4.70	0.030
City	1426(38.3)	100.3	86.0~119.2		
Rural	2301(61.7)	98.8	83.2~118.6		
Location				1.72	0.423
eastern	1280(34.3)	100.2	84.9~118.4		
midlands	1133(30.4)	99.4	85.5~119.6		
western	1314(35.3)	98.6	83.0~118.6		
Race*				13.15	<0.001
Han	3317(89.0)	99.8	85.0~119.3		
Minority	410(11.0)	95.6	79.5~113.6		
Education level				10.40	0.006
elementary school and below	2820(75.7)	98.7 ^a^	83.4~118.2		
middle school	810(21.7)	101.8 ^b^	87.6~120.6		
college degree and above	97(2.6)	99.8 ^ab^	86.5~121.1		
Family income per annum (10,000 yuan)				0.01	1.000
<1	1732(46.5)	99.7	83.9~119.7		
1~2.5	465(12.5)	99.0	84.7~116.8		
2.5~ 5	762(20.4)	99.5	84.2~118.0		
>5	768(20.6)	98.6	84.7~114.7		

^a,b^ Horizontal comparison of differences between groups; the same letter has no difference, different letters have differences. BMI, body mass index; N, number; IQR, interquartile range. * Han, largest nationality in China, accounting for 90% of China’s total population; Minority, Except for the Han nationality, the other 55 legal nationalities are ethnic minorities.

**Table 2 nutrients-13-03086-t002:** Prevalence of zinc deficiency and multivariate logistic regression analysis in Chinese elderly in CACDNS 2015.

Variables	Zinc Deficiency *			OR ^#^	95% CI	*p* Value
	Prevalence (%)	95% CI(%)	*p* Value			
Total population	8.68	7.74~9.61		–	–	–
Sex			<0.001			
Man	11.23	9.73~12.73		1.85	1.47~2.34	<0.001
Woman	6.22	5.10~7.34		–	–	–
Age group (Years)			0.002			
60~69	7.34	6.07~8.61		–	–	–
≥70	10.39	9.02~11.75		1.44	1.14~1.82	0.00
BMI(kg/m^2^)			<0.001			
Low weight (<18.5)	14.22 ^a^	9.63~18.81		1.33	0.91~1.94	0.14
Normal (18.5~<24.0)	10.32 ^a^	8.89~11.75		–	–	–
Overweight (24.0~<28.0)	6.33 ^b^	4.91~7.76		0.67	0.51~0.88	0.00
Obesity(≥28.0)	5.75 ^b^	3.43~8.07		0.52	0.33~0.80	0.00
Anemia			0.411			
No	8.56	7.57~9.54		–	–	–
Yes	9.85	6.77~12.92		1.01	0.69~1.46	0.98
Residence region			0.015			
City	7.32	5.89~8.74		–	–	–
Rural	9.75	8.51~10.98		1.28	0.99~1.65	0.06
Location			0.350			
Eastern	8.02	6.48~9.55		–	–	–
Midlands	8.38	6.71~10.04		0.99	0.75~1.31	0.94
Western	9.60	7.95~11.25		1.07	0.82~1.40	0.63
Race			0.006			
Han	8.22	7.26~9.18		–	–	–
Minority	12.46	9.08~15.84		1.39	1.01~1.91	0.04
Education level			0.199			
Elementary school and below	9.07	7.97~10.16		–	–	–
Middle school	8.00	6.06~9.93		1.01	0.75~1.35	0.95
College degree and above	4.44	0.54~8.33		0.80	0.34~1.87	0.60
Family income per annum (10,000 yuan)			0.034			
<1	9.64 ^a^	8.26~11.02		–	–	–
1~2.5	9.35 ^a^	7.24~11.47		1.01	0.76~1.33	0.95
2.5~5	7.66	4.93~10.39		0.81	0.54~1.20	0.29
>5	5.94 ^b^	4.09~7.78		0.78	0.54~1.11	0.17

^a,b^ Horizontal comparison of differences between groups; the same letter has no difference, different letters have differences.CI, confidence interval; OR, odd ratio; CI, confidence interval. * Serum zinc deficiency: elderly men < 74 μg/dL, elderly women < 70 μg/dL. # Adjusted for age, BMI, residence region.

## Data Availability

Not applicable.

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
