# Peer review of "Zinc Nutritional Status and Risk Factors of Elderly in the China Adult Chronic Disease and Nutrition Surveillance 2015"

_nutrients, 2021, doi:10.3390/nu13093086_

Round 1

Reviewer 1 Report

The manuscript titled "Zinc nutritional status and risk factors of elderly in the China Adult Chronic Disease and Nutrition Surveillance 2015" promises interesting insights into the zinc supply situation of the elderly population in China. The strength of the study is the number of people included and the multicentre analysis. However, the current script still has major deficits that need to be revised. Particularly the current version of the English is not adequate enough for a sufficient review. Especially compared to the previous study "Evaluation of Serum Zinc Status of Pregnant Women in the China Adult Chronic Disease and Nutrition Surveillance (CACDNS) 2015", there are substantial differences in the quality of the language presentation.

Examples:

“Collected blood samples from subjects and the basic information were also collected. According to the criteria of serum zinc deficiency recommended by International Zinc Nutrition Consulting Group (IZiNCG), and further analyzed its influencing factors.”

“Before the analyses, put the frozen serum at room temperature to thaw naturally, draw 100 μL of the serum into a 15 ml centrifuge tube, weigh w0, dilute to 2 mL with 0.5% 80 (v/v) dilute nitric acid, weigh w1 and mix well.”

Furthermore, the methods need to be described in more detail.

From how many individuals were the 3727 people finally selected? How many subjects were excluded? How were "abnormal detections" defined? How were outliers excluded? How was the data normality/abnormality examined?

The table needs to be revised. From the BMI listing onwards, the values are not correctly assigned. 

The discussion should also be revised. The results are presented again without discussing them with previous findings in the literature. Statements like the following should be discussed in more detail. “As we have mentioned above, zinc deficiency is harmful to the immunity system. And the relatively higher prevalence of zinc deficiency in elder might make them more susceptible to other diseases. So more attention should be paid to this population.” For example: What conclusions should be drawn from this? What does this potentially mean for the spread of infectious diseases within a whole population or specific population groups? Are there studies on specific diseases in this age group, such as zinc deficiency and mortality in COVID-19? Does the identification of a prevalent deficit in this age group offer a potential opportunity for intervention?

Reviewer 2 Report

Dear Authors ,

Your research design is appropriate and interesting. But some of the sentences are very difficult to understand. Moreover there are many faults in Table 1. That is one of the reason your  conclusions cannot be understood.

This manuscript needs major issues, particularly in Table 1. It seems that there is errors in the alignment of the lines of the columns. For example, in the line BMI you have 533 (14,3 %) and nothing in the line low weight. The same for race, residence, region, et cetera.

More important in the case of race if you add up the Han (1314 ??? ) and the minority (3317) the total number is greater than the total population (3727 ).

Therefore it is impossible to understand the results in Table 1 and accordingly in Table 2, thereby the results.

A revision of the English text should be done. For example, sentence line 172 is long and not clear. In the abstract, three sentences, line 16, line 22 and line 28 are not easy to understand from my point of view.

As there are many faults, it is not possible to point all the minor issues . My suggestion is a first revision with Table 1 corrected and a revision of the English , and then minor issue may be corrected if necessary.

Best regards

Round 2

Reviewer 1 Report

I would like to thank the authors for revising the manuscript. The manuscript has improved in quality and now allows the reader a better understanding of the results of this study. Nevertheless, I would recommend the authors to make some minor linguistic and content adjustments.

"The serum zinc concentration of the elderly is related to gender, age group, BMI, residence region, race and education level. Serum zinc concentration in elderly men is higher than that in elderly women (p < 0.05)..." "that" should be deleted

"The serum zinc concentration can well reflect the nutritional supplement effect of dietary zinc and zinc supplements, and is suitable for evaluating the zinc nutritional status of men and women at all agesall the populations." "the" is not necessary

"Therefore, it is still necessary to continue to paid more attention to this population". "to pay" should be the right tense

"In this our study, we also compared the zinc deficiency rate between anemia subjects with the normal ones. Tthe result should showed thatalthough the elderly with anemia showed a slightly higher zinc deficiency rate(9.85 vs 8.56), but the difference was not statistical significant (p > 0.05). Moreover, result of the logistic regression analysis also showed the anemia was not the risk factor for zinc deficiency in the elderly of our study (OR=1.01, 95%CI 0.69-1.46). More research is warranted in this field." Just a stylistic note.

In this paragraph and in the preceding paragraphs, the sentences are repeatedly opened with "our results showed" and the presentation of results is also introduced with "showed". In this paragraph alone three times. Synonyms such as "to reveal", "to yield" or "to exhibit" would be conceivable alternatives.

Furthermore:

1.In this scientific context, I would suggest to use the term "sex" for the distinction between men and women. "Gender" rather corresponds to a societal definition or an individual identification with the respective sex.

2. For readers who are not so well acquainted with China's ethnic groups, an explanation of the division into Han vs non-Han would be helpful. This would facilitate understanding in the following when "minorities" are mentioned. Furthermore, a short explanation below the tables would also be advantageous in this respect. 

3. "This study found that ageaged above 70y, gendermale, and minoritiesethnicity were risk factors for zinc deficiency. Generally, the physiological requirement of zinc in male is higher than female. And the Dietary Reference Intakes (DRIs) in China [19] also recommended a higher dietary reference intake for male than for female. In our study, the male elderly was in higher risk than female might be related to this. The results of this study show that the risk of zinc deficiency is higher for elderly over 70y compared to 60-69y, this might be due to increase with age, food intake, chewing ability and decreased ability to digest."

I would recommend revising this paragraph. The opening is well chosen. In the first sentence, the identified risk factors for zinc deficiency are mentioned. In the following, possible reasons for zinc deficiency are mentioned. The explanations for the difference between men and women seem understandable. But then it does not need to be presented again that people over 70 have an increased risk. This has already been described in the first sentence. Furthermore, the argumentation "this could be due to the increase with age..." should be language revised. There is no statement on the last risk factor. Are there studies on why belonging to an ethnic minority could be a risk factor?

Reviewer 2 Report

Dear Authors,

Thank you for revising your manuscript.

But there are still errors in Table 1.

It seems that there are still errors in the alignment of the lines of the column but not only. For example, where are the western patients? You have 1426 (38,3 %) patients of the eastern, and 2301 (61, 7 % ) patients of the midlands, that give us the total number of patients ( 3727 ). In table 2 the three lines Eastern, Midlands and Western are present. How do you explain the discrepancy between the two tables?

In race there are three lines: race 1280 , Han ( 1133 ) and minority ( 1314 ) . That give us a total of 3727. In Table 2 there are two entities of race, the Han ethnicity and the minority? How again do you explain the discrepancy between the two tables?

The line Family income is very difficult to understand.

Therefore it is still impossible to approve the results in Table 1 and accordingly in Table 2, thereby the results.

I have a remark in materials and methods:

Line 65: 14 elderly patients were selected. On which basis were they selected?

A revision of the English text should still be done concerning the sentences that were added. For example, sentence line 67 is long and not clear. No verb in line 79. There are sometimes time errors and in line 93 it should be written the serum zinc concentration was (and not were) et cetera.

The end of the abstract is still not clear beginning from line 22 and must be revised.

Best regards
